# Measuring the tolerance of the genetic code to altered codon size

**Erika Alden DeBenedictis[1,2]\*, Dieter Söll[3], Kevin M Esvelt[2]**

[1]Department of Biological Engineering, Massachusetts Institute of Technology, Cambridge, United States; [2]Department of Media Arts and Sciences, Massachusetts Institute of Technology, Cambridge, United States; [3]Department of Molecular Biophysics and Biochemistry, Yale University, New Haven, United States

**Abstract** Translation using four-base codons occurs in both natural and synthetic systems. What constraints contributed to the universal adoption of a triplet codon, rather than quadruplet codon, genetic code? Here, we investigate the tolerance of the *Escherichia coli* genetic code to tRNA mutations that increase codon size. We found that tRNAs from all 20 canonical isoacceptor classes can be converted to functional quadruplet tRNAs (qtRNAs). Many of these selectively incorporate a single amino acid in response to a specified four-base codon, as confirmed with mass spectrometry. However, efficient quadruplet codon translation often requires multiple tRNA mutations. Moreover, while tRNAs were largely amenable to quadruplet conversion, only nine of the twenty aminoacyl tRNA synthetases tolerate quadruplet anticodons. These may constitute a functional and mutually orthogonal set, but one that sharply limits the chemical alphabet available to a nascent all-quadruplet code. Our results suggest that the triplet codon code was selected because it is simpler and sufficient, not because a quadruplet codon code is unachievable. These data provide a blueprint for synthetic biologists to deliberately engineer an all-quadruplet expanded genetic code.

## Editor's evaluation

Using a phage-based library generation and selection, the authors generated a suite of 4-base decoding tRNAs with improved efficiency in quadruplet decoding. The data represent an important step toward enhancing protein synthesis with 4-base codons. Overall, the approach to generate many tRNA variants with quadruplet anticodons is intriguing and provides a wealth of valuable information to the field. The results should become foundational for the field of synthetic biology.

**\*For correspondence:** erika.alden@mit.edu

## Introduction

The genetic code is determined by a combination of tRNAs and aminoacyl tRNA synthetases (AARSs). Codons are dictated by the three bases in the center of the anticodon loop of each tRNA, which undergo Watson-Crick base pairing to an mRNA transcript during translation, enabling accurate codon recognition. The correspondence between codons and amino acids – one tRNA isoacceptor class for each canonical amino acid – is dictated by the 20 AARSs which specifically recognize bases (identity elements) in the tRNAs, and attach the cognate amino acid onto only the CCA 3' terminus of the cognate tRNA. The aminoacylation process is exquisitely accurate, enabling high-fidelity protein synthesis (*Reynolds et al., 2010*). Anticodon mutations frequently alter or abolish selective charging with the cognate amino acid because most AARSs rely on bases in the anticodon to identify the cognate tRNA (*Giegé et al., 1998*). However, certain natural anticodon mutations generate 'suppressor' tRNAs that insert their cognate amino acid in response to 5'-UAG-3' stop codons pairing with its 5'-CUA-3' anticodon (*Eggertsson and Söll, 1988*). Frameshift suppression, in which quadruplet

**Figure 1.** Evolution of quadrupelt tRNAs.
We studied whether tRNAs can arise through simple changes to the anticodon followed by additional mutations accumulated during evolution.

tRNAs (qtRNAs) 'suppress' a +1 frameshift mutation can also arise (*Riddle and Carbon, 1973*; *Roth, 1981*); that is, qtRNA-Gly-GGGG can decode the four-base codon 5'-GGGG-3' in mRNA transcripts using the 5'-CCCC-3' anticodon. In the presence of efficiently aminoacylated qtRNAs, the ribosome is capable of translation with individual non-canonical stop or quadruplet codons within an otherwise all-triplet transcript (*Dunkelmann et al., 2020*; *de la Torre and Chin, 2021*). If individual quadruplet codon translation is known to arise through simple point insertions, what functional constraints, if any, prevent the natural or synthetic evolution of an all-quadruplet genetic code?

These origin-of-life questions have newfound importance to engineering with the advent of genetic code expansion technology. An expanded all-quadruplet genetic code would offer 256 total codons (*de la Torre and Chin, 2021*), including hundreds of free codons that could be assigned to non-canonical amino acids (ncAAs), valuable chemical additions to the genetic code that enable improved protein therapeutics (*Hutchins et al., 2011*). Indeed, the efficiency of quadruplet translation can be so high that as many as four unique quadruplets can be translated within one transcript to incorporate natural (*DeBenedictis et al., 2021*) or non-canonical (*Dunkelmann et al., 2021*) amino acids site-specifically. These inspiring results raise the question: is it possible to create an all-quadruplet genetic code? Such a code would presumably require the ability to incorporate the 20 canonical amino acids directed by quadruplet codons. Might the necessary translation components be easily derived from existing tRNAs and AARSs?

To investigate these questions, we present a comprehensive study of whether tRNAs in general (from all 20 isoacceptor classes) can decode diverse quadruplet codons in *Escherichia coli* (we examined 57 of the 256 possible quadruplet codons). We systematically explored whether quadruplet codon translation can arise through simple point insertions in each of the tRNA anticodon loops, or through mutation of many bases in the anticodon. We then used directed evolution to determine how often additional mutations throughout the tRNA can improve translation of the resulting qtRNAs (*Figure 1*). Finally, we characterized the fidelity quadruplet codon translation. Remarkably, we found that 12/20 isoacceptor classes of tRNAs can be readily converted to selectively charged qtRNAs, as confirmed with mass spectrometry. The efficiency of quadruplet decoding is often low, but can frequently be improved by accumulating additional mutations along the sides of the anticodon loop. Most of the resulting qtRNAs selectively incorporate a single amino acid in response to a quadruplet codon. Our results identify some of the barriers limiting the adoption of quadruplet codons by natural evolution, and present 9/20 qtRNAs necessary to synthetically create an all-quadruplet expanded genetic code.

## Results
### Evolution of qtRNAs through point insertions

Many known examples of frameshift suppressors contain single base insertions in the anticodon that convert a triplet tRNA into a qtRNA. We initially tested whether tRNAs can evolve into qtRNAs through simple point insertions. We selected 21 endogenous *E. coli* tRNAs, one cognate tRNA for each canonical amino acid and the initiator methionine tRNA (*Supplementary file 1*, Materials and

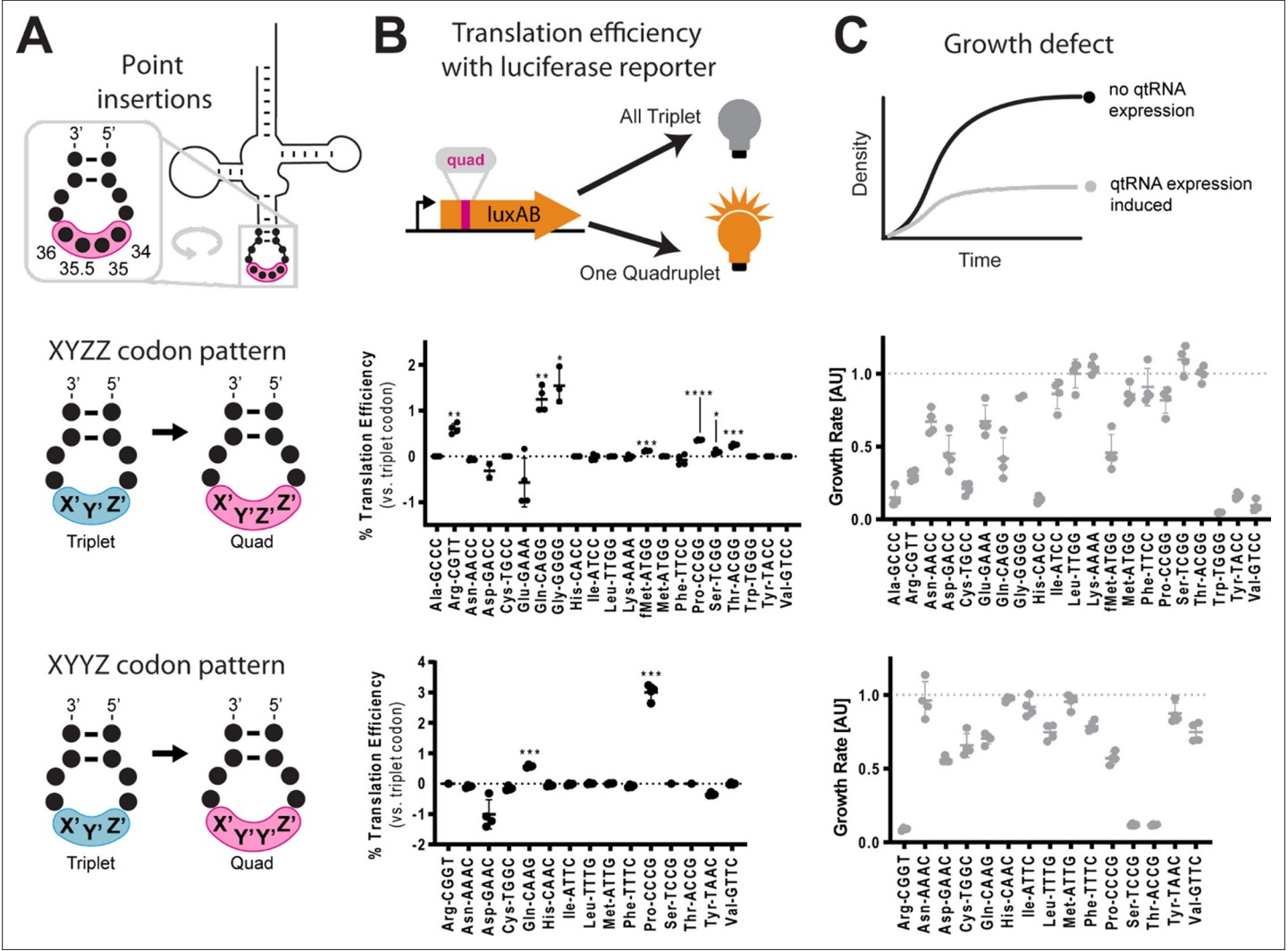

**Figure 2.** Engineering qtRNAs with codon patterns. (A) We measured quadruplet tRNAs (qtRNAs) that might arise through point insertions in the anticodon loop. Each qtRNA is based upon a tRNA from the *Escherichia coli* genome that serves as a 'scaffold' (*Supplementary file 1*). We tested two quadruplet codon patterns: a tRNA decoding the triplet codon 'XYZ' to a qtRNA decoding the quadruplet codon 'XYZZ' or 'XYYZ'. In instances in which XYZZ and XYYZ are the same, the qtRNA is depicted on the XYZZ graph. We use 'qtRNA'-'three letter scaffold'-'four letter codon' nomenclature to refer to qtRNAs; for example, a serine qtRNA bearing a 5'-UCUA-3' anticodon that recognizes 5'-UAGA-3' in mRNA transcripts is referred to as qtRNA$^{Ser}_{TAGA}$. (**B**) We measured qtRNAs using a luciferase readthrough assay. Measurements are taken kinetically and normalized to culture density, and efficiency is reported relative to luminescence produced by a wildtype (WT), all triplet luciferase transcript. qtRNAs that are statistically >0 are annotated with their one-sample t-test p-value: 0.033(*), 0.0021(**), 0.0002(***), < 0.001(****). fMet qtRNAs are measured with a luciferase reporter bearing a quadruplet codon at residue 1; all others are measured with a quadruplet codon at residue 357 of luxAB. (**C**) Expression of qtRNAs can be toxic. Here, we report the fractional OD$_{600}$ density difference between cultures where qtRNA expression had been induced versus suppressed. Data in (**B** and **C**) represent the mean and standard deviation of three to eight technical replicates in one biological replicate. For raw data, see *Figure 2—source data 1*.

The online version of this article includes the following source data for figure 2:

**Source data 1.** Luminescence and growth metric of XYZZ and XYYZ qtRNAs.

methods – 'tRNA scaffold selection'), to serve as 'scaffolds' into which we introduced anticodon point insertions. We tested two point insertion locations: if the original tRNA decodes the triplet codon 'XYZ', we created a qtRNA that decodes 'XYZZ' and a qtRNA that decodes 'XYYZ' (*Figure 2A*). These anticodon patterns preserve the nature of the bases in the anticodon loop, which most AARSs use to recognize the cognate tRNA (*Giegé et al., 1998*), and are found in known qtRNAs, such as sufD (*Riddle and Carbon, 1973*) (qtRNA$^{Gly}_{GGGG}$) and sufG (*O'Connor, 2002*) (qtRNA$^{Gln}_{CAAA}$). Throughout this paper, for ease of comparison to standard triplet codon tables, we use qtRNA$^{three\ letter\ scaffold}_{four\ letter\ DNA}$

$_{codon}$ nomenclature to refer to qtRNAs; for example, a serine qtRNA bearing a 5'-UCUA-3' anticodon is referred to as qtRNA$^{Ser}_{TAGA}$.

We used two techniques to characterize these qtRNAs. First, to measure the quadruplet codon translation efficiency, we used a luciferase readthrough assay (Materials and methods – 'Luciferase readthrough assay'). This reporter contains a single quadruplet codon at permissive residue 357 of *luxAB* (*DeBenedictis et al., 2021*); failure to decode the quadruplet codon leads to premature termination, whereas successful four-base decoding results in full-length luxAB translation and luminescence (*Figure 2B*). Seven of the twenty 'XYZZ'-decoding qtRNAs (qtRNA$^{Arg}_{CGTT}$, qtRNA$^{Gln}_{CAGG}$, qtRNA$^{Gly}_{GGGG}$, qtRNA$^{fMet}_{ATGG}$, qtRNA$^{Pro}_{CCGG}$, qtRNA$^{Ser}_{TCGG}$, qtRNA$^{Thr}_{ACGG}$) and two of the 'XYYZ'-decoding qtRNAs (qtRNA$^{Gln}_{CAAG}$, qtRNA$^{Pro}_{CCCG}$) functionally decode a quadruplet codon during translation. The frequent functionality of the XYZZ codon pattern may be due to flexibility in synthetase recognition at the third position of the codon, which is frequently a wobble base pair. Next, we quantified the toxicity of qtRNA expression by comparing the growth defect (Materials and methods – 'Growth defect') of cultures with qtRNA expression induced or suppressed (*Figure 2C*). The qtRNAs fall into several categories: those that exhibit no fitness defect such as the naturally occurring and highly functional qtRNA$^{Gly}_{GGGG}$; those that exhibit severe fitness defects and effectively halt bacterial growth upon induction such as qtRNA$^{Ala}_{GCCC}$, and those that moderately slow bacterial growth. The translation efficiency and growth defect of qtRNAs depends upon more than just the interaction with the cognate AARS: AlaRS, LeuRS, and SerRS are all tRNA synthetases that do not interact with the anticodon loop of their cognate tRNA, yet qtRNAs derived from these scaffolds exhibit a range of behaviors depending upon the new anticodon. For example, qtRNA$^{Ser}_{TCCG}$ exhibits high growth defect, while qtRNA$^{Ser}_{TCGG}$ exhibits no growth defect and modest quadruplet translation efficiency. Together, these data show that a third of the 20 isoacceptor classes have access to single base insertions that enable modestly functional quadruplet codon translation, however, other point insertions create qtRNAs that do not functionally decode quadruplet codons or incur large growth defects when expressed.

## Evolution of qtRNAs through anticodon replacement

Next, we tested whether tRNAs could evolve into qtRNAs through whole anticodon replacement, as might occur during recombination or more intense mutagenesis. Antibiotic selection markers have previously been used to identify functional qtRNAs (*Magliery et al., 2001*). We applied an equivalent approach based on the use of an M13 bacteriophage tail fiber pIII as a selection marker (*DeBenedictis et al., 2022*). In this selection scheme, a qtRNA is encoded on the genome of a ΔpIII M13 bacteriophage. Phage are challenged to infect bacteria bearing a plasmid that encodes pIII containing a quadruplet codon at permissive residue 29 (*Bryson et al., 2017*). Functional qtRNAs are capable of producing full-length pIII and thus phage progeny, while nonfunctional qtRNAs result in production of truncated pIII and thus no further phage.

For each of the 20 representative *E. coli* tRNA scaffolds used above, we created a 256-member qtRNA library containing degenerate anticodons (*Figure 3A*, Materials and methods – 'Phage library primer design', 'Phage library cloning'). We selected eight quadruplet codons of interest, focusing on codons for which at least one functional qtRNA was already known that could act as positive control. We selected for functional qtRNAs from these libraries (*Figure 3B*, Materials and methods – 'pIII-based selection of NNNN anticodon libraries') to identify qtRNAs that decode the eight quadruplet codons of interest (*Figure 3C*). High final phage titers after selection indicate the presence of a functional qtRNA, and we selected 69 putative qtRNAs that exhibit high final phage titer for further characterization (*Figure 3D*). We used plaque assays (Materials and methods – 'Phage plaque assays') to isolate clonal phage variants and determined the anticodon of each highly selected variant with Sanger sequencing. We found that for most variants, the qtRNA agrees with the quadruplet codon in the reporter at all four positions, or the first three positions of the codon (*Figure 3E*), in agreement with previous findings on quadruplet codon crosstalk with fourth-base mismatches (*Anderson et al., 2002*). We found that qtRNAs identified by this phage-based assay are also functional in the luciferase assay. These results demonstrate that most tRNA scaffolds are capable of supporting quadruplet codon translation through whole anticodon replacement.

In addition to the expected anticodons, we found that some qtRNAs instead matched quadruplet codons that appear nearby in the sequence context of the reporter (*Figure 3F*). In these cases, qtRNAs were validated using this novel codon in position 357 of the luciferase reporter, confirming that they

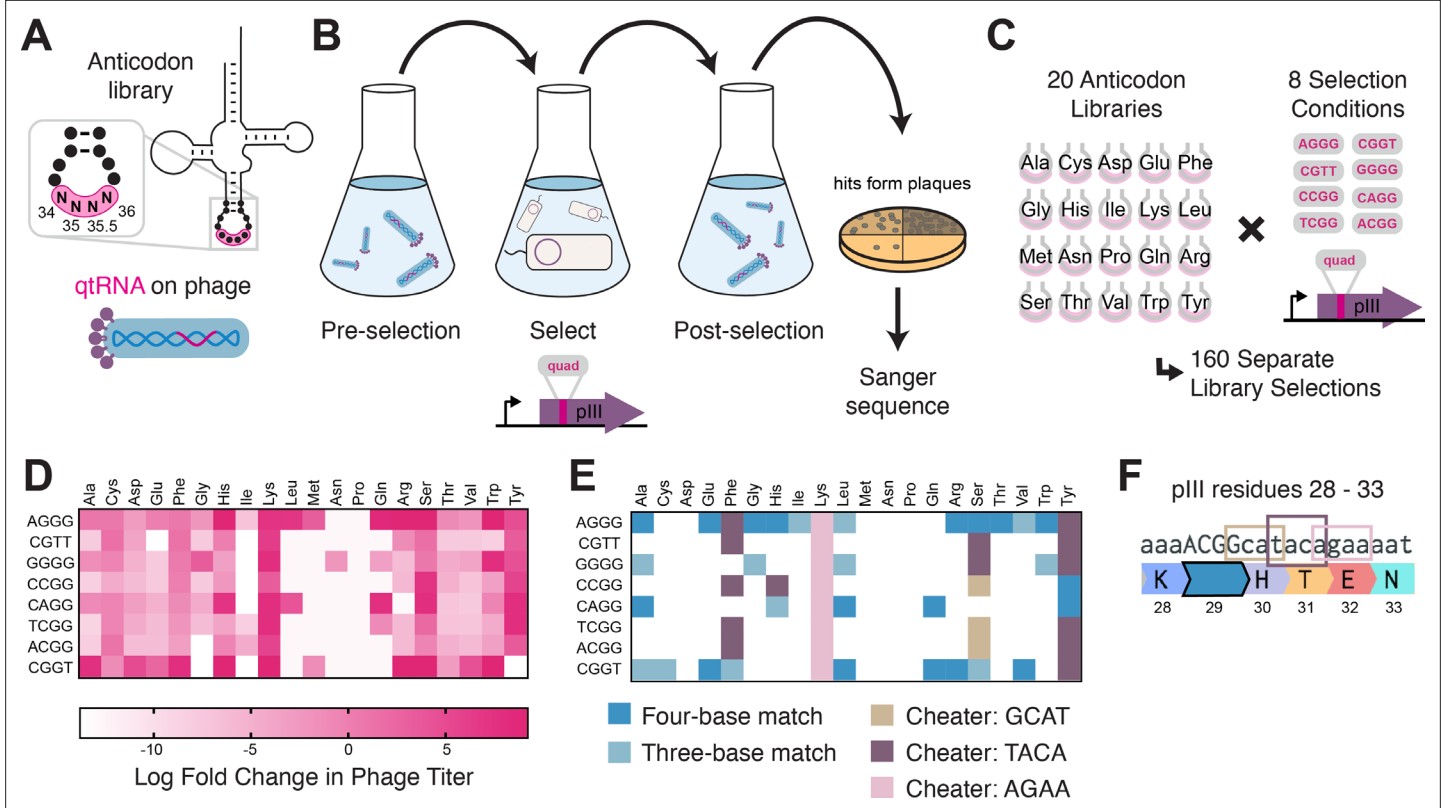

**Figure 3.** Selection for functional qtRNAs. (**A**) We created quadruplet tRNAs (qtRNA) libraries using degenerate primers to randomize the four bases in the anticodon. The qtRNA library is expressed from a ΔpIII M13 bacteriophage. (**B**) We selected these libraries by challenging the phage to infect and propagate in bacteria that require a quadruplet codon to be translated in order to produce a functional version of the essential phage gene, pIII. Plaques from selected libraries were Sanger sequenced. (**C**) We crossed each library with each pIII-based reporter for a total of 160 separate library selections. (**D**) The log fold change in phage titer when comparing the post-selection population to the pre-selection population. Data represent the mean in one biological replicate. For raw data, see *Figure 3—source data 1*. (**E**) For every filled square, we sequenced two plaques in order to determine the anticodon identity. Results include instances in which the qtRNAs match the reporter at all four positions, or that match with the first three bases of the codon (shades of blue). Additionally, we show instances in which qtRNAs were discovered that suppress a different quadruplet codon near residue 29. (**F**) Although we intend for the phage to suppress the quadruplet codon located at permissive residue 29 of pIII (*Bryson et al., 2017*), in several instances the selection identified qtRNAs that suppress a nearby quadruplet codon instead.

The online version of this article includes the following source data and figure supplement(s) for figure 3:

**Figure supplement 1.** Codon reassignment for asparagine and methionine.

**Source data 1.** Log fold change in phage titer.

decode the novel codon in two different sequence contexts. We noticed that the novel codons often bear similarity to the qtRNA's original codon; that is, qtRNA$^{Lys}_{AGAA}$ is highly similar to the scaffold's original AAA codon, and qtRNA$^{Tyr}_{TACA}$ and qtRNA$^{Phe}_{TACA}$ are similar to TAC and TTC, respectively. The emergence of these anticodons suggests that these isoacceptor classes favor quadruplet codons that are related to their natural triplet codon, and demonstrates that qtRNA evolution depends upon the sequence context of relevant ORFs.

Together, these experiments identified functional qtRNAs involving four or fewer mutations for 18/20 isoacceptor classes. For the remaining two isoacceptor classes, Met and Asn, we systematically tested additional codons and found that qtRNA$^{Met}_{AGGG}$ and qtRNA$^{Asn}_{AGGA}$ both exhibit weak quadruplet codon translation (*Figure 3—figure supplement 1*). Therefore, every isoacceptor class can give rise to qtRNAs capable of decoding quadruplet codons during protein translation.

## Directed evolution of qtRNAs

Having found that qtRNAs that functionally decode quadruplet codons can arise generally through just a few mutations, we sought to understand other factors that may prevent more widespread use

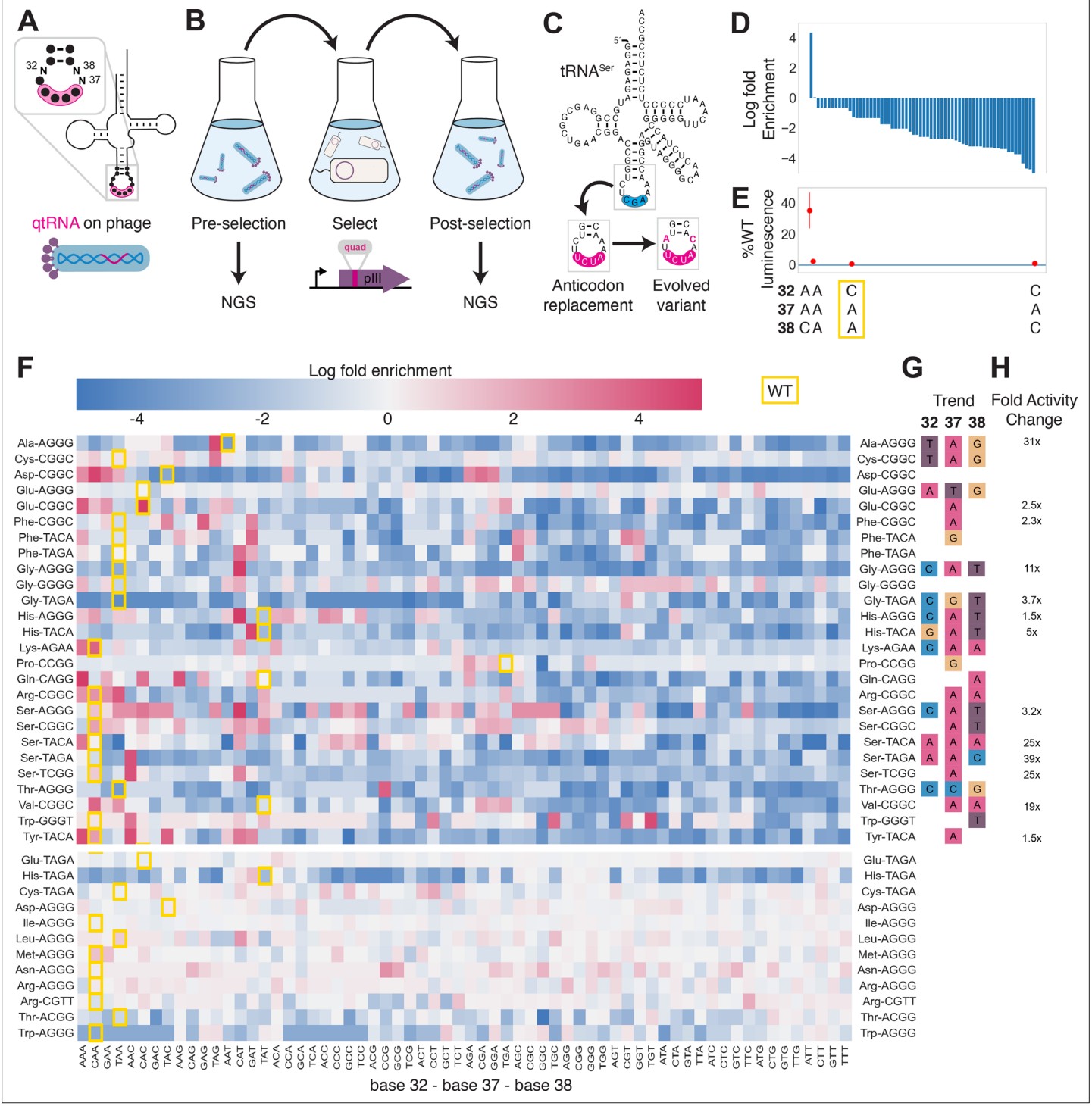

**Figure 4.** Directed evolution of anticodon loop sides. (**A**) We created quadruplet tRNAs (qtRNA) libraries using degenerate primers to randomize positions 32, 37, and 38 of the anticodon. The qtRNA library is expressed from a ΔpIII M13 bacteriophage. (**B**) We selected these libraries by challenging the phage to infect and propagate in bacteria that require a quadruplet codon to be translated in order to produce a functional version of the essential phage gene, pIII. Libraries were next-generation sequencing (NGS) sequenced to >10× library size before and after selection. (**C**) tRNA$^{Ser}_{TCG}$ is known to be a scaffold for the functional qtRNA$^{Ser}_{TAGA}$ after anticodon replacement alone. Additional mutations to the sides of the anticodon loop are known to improve quadruplet codon translation efficiency. (**D**) Log fold enrichment of the population abundance and (**E**) translation efficiency as measured by a luciferase readthrough assay of the 64 possible combinations of nucleotides at positions 32, 37, and 38. (**F**) Log fold enrichment of all 41 qtRNA libraries for each of the 64 library members. Libraries are separated by those that exhibit abundance changes during selection (above) from those that exhibit no significant abundance changes (below). The anticodon loop sides present in the wildtype (WT) tRNA scaffold are boxed in gold. For raw data,

*Figure 4 continued on next page*

*Figure 4 continued*

see *Figure 4—source data 1*. (**G**) For each library, the trend in nucleotide preference for each position is listed. (**H**) Nucleotide preferences for select libraries were measured by cloning a qtRNA variant and measuring it using a luciferase readthrough assay. The fold improvement in activity over the WT values of base 32, 37, and 38 are listed. In (**e and f**), the original identities of bases 32, 37, and 38 found in the WT triplet tRNA scaffold are boxed in gold.

The online version of this article includes the following source data for figure 4:

**Source data 1.** Log fold enrichment of anticodon loop side libraries.

of quadruplet codons. Many qtRNAs were quite inefficient in translation: the presence of a single quadruplet codon in an mRNA transcript can reduce total protein yield to less than 3% relative to an all-triplet mRNA (*Figure 2*). Mutations at the anticodon loop sides of the qtRNAs have been observed to improve translation efficiency for TAGA-qtRNAs (*DeBenedictis et al., 2021*; *DeBenedictis et al., 2022*; *Niu et al., 2013*). Triplet tRNAs are known to exhibit patterns that relate the bases in the anticodon loop sides to the bases in the anticodon itself (*Yarus, 1982*), and similarly benefit from anticodon loop side mutations after anticodon replacement (*Kleina et al., 1990*; *Raftery and Yarus, 1987*; *Cervettini et al., 2020*). In some cases, mutations in this area can alter qtRNA charging; in others they improve quadruplet translation efficiency without altering the qtRNA's interaction with the cognate AARS (*DeBenedictis et al., 2021*). We hypothesized that qtRNAs in general require mutations at bases 32, 37, and 38 to better accommodate a new codon, and that this requirement may present a key barrier preventing the natural evolution of efficient qtRNAs.

To test this hypothesis experimentally, we selected 41 functional qtRNAs, including at least one qtRNA for every unique scaffold, and cloned a library containing degenerate nucleotides at bases 32, 37, and 38 (*Figure 4A*, Materials and methods – 'Phage library primer design', 'Phage library cloning'). We selected functional members of these libraries using the pIII-based selection and used next-generation sequencing (NGS) to characterize the abundance of each library member before and after selection (*Figure 4B*, Materials and methods – 'pIII-based selection and NGS of libraries diversified at positions 32, 37, and 38').

We began by assessing results for the well-studied qtRNA$^{Ser}_{TAGA}$, which is known to have an improved variant, qtRNA$^{Ser}_{TAGA}$-32A-38C, that exhibits improved quadruplet codon translation but unaltered, selective aminoacylation with serine (*DeBenedictis et al., 2021*; *Figure 4C*). Of the 64 possible combinations of DNA bases at positions 32, 37, and 38, the single library member A32 A37 C38 is enriched four log fold above all other variants (*Figure 4D*). We measured several qtRNA$^{Ser}_{TAGA}$ variants that correspond to different levels of enrichment using a luciferase readthrough assay, and confirmed that the strongly enriched variant exhibits more efficient quadruplet decoding than deenriched variants (*Figure 4E*).

Next, we applied the same procedure to quantify fold enrichment for the other 40 qtRNA libraries (*Figure 4F*). We identified 15 libraries that exhibit no selective pressure, 12 libraries that strongly enrich a single library member, like qtRNA$^{Ser}_{TAGA}$, and an additional 11 libraries exhibit strong enrichment for library members with a specific base at one or two positions, but not all three (*Figure 4G*). For several of these libraries, we used a luciferase readthrough assay to measure the fold change activity when mutations to bases 32, 37, and 38 are introduced. In most cases, introduction of these mutations substantially increases quadruplet codon translation efficiency (*Figure 4H*).

We were curious whether there are overall trends in the identity of optimal anticodon loop sides for efficient quadruplet codon translation. The optimal library member is not determined by the scaffold or codon independently, indicating that the mechanism by which these mutations improve quadruplet codon translation does not improve the qtRNA's interaction with its respective AARS. Among the libraries there was a prominent preference for A37, a base known to be associated with reading frame maintenance (*Agris, 2004*). Additionally, C32 A37 T38 and T32 A37 G38 appear often among libraries that exhibit strong preference for a single library member. The presence of modified nucleotides is especially important at two sites: position 34, the first base of the anticodon, and position 37, the nucleotide downstream of the anticodon (*Agris, 2004*; *Grosjean and Westhof, 2016*). The location of 'identity elements' for some of these modifying enzymes within the anticodon loop sides has been implied (*Grosjean and Westhof, 2016*). Mutation of these bases may improve the RNA modification of the anticodon loop, which is essential for many tRNA functions (*Edwards et al., 2020*).

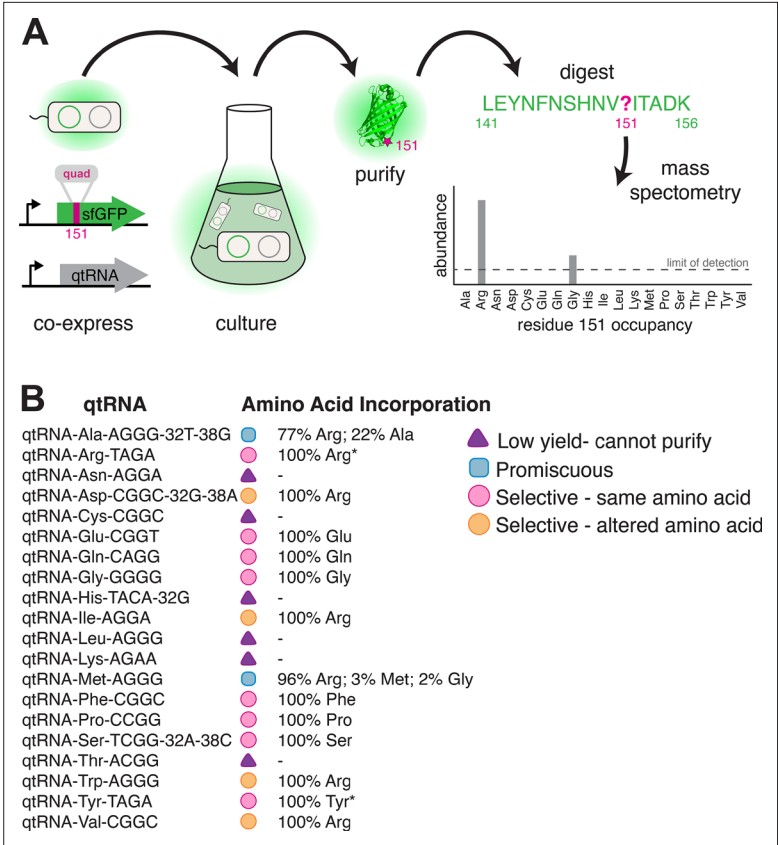

**Figure 5.** Characterization of amino acid incorporation by quadruplet tRNAs (qtRNAs). (**A**) We characterized the amino acid incorporated during translation by co-expressing a qtRNA and an sfGFP-151-quad transcript, purifying the resulting GFP, and analyzing the occupancy of residue 151 using mass spectrometry (*Supplementary file 3*). (**B**) Results of applying this pipeline to at least one qtRNA based on each of the 20 canonical scaffolds (Materials and methods – 'Quantification of qtRNA charging using mass spectrometry'). Charging of (*) qtRNA$^{Arg}_{TAGA}$ and qtRNA$^{Tyr}_{TAGA}$ has been previously reported (*DeBenedictis et al., 2021*). Data represents the mean of one biological replicate. For some qtRNAs, yield was too low to allow for charging characterization even when purified at 1 L scale (*Supplementary file 2*). Raw spectra have been deposited in the PRIDE database (*Perez-Riverol et al., 2022*), dataset identifier PXD031925 and 10.6019/PXD031925.

## Which amino acid(s) do qtRNAs incorporate during translation?

Having established that anticodon loop changes can often produce tRNAs that decode four-base codons, we sought to determine which amino acid these qtRNAs incorporate during translation. To do so, we translated *sfGFP* mRNA containing a quadruplet codon at permissive residue 151 (*Young et al., 2010*) in the presence of a qtRNA. We then used mass spectral analysis to determine the nature and occupancy of amino acid 151 in the resulting protein (*Cervettini et al., 2020*) for qtRNAs based on the 20 distinct tRNA scaffolds (*Figure 5*, Materials and methods – 'Quantification of qtRNA charging using mass spectrometry', *Supplementary file 3*).

We were unable to characterize the acylation properties of six qtRNAs due to low sfGFP purification yield, even when purified at 1 L scale (*Supplementary file 2*) (qtRNA$^{Asn}$, qtRNA$^{Cys}$, qtRNA$^{His}$, qtRNA$^{Leu}$, qtRNA$^{Lys}$, and qtRNA$^{Thr}$). These qtRNAs would be unlikely to be selected in a natural system due to their inability to produce full-length protein efficiently. Eight qtRNAs are selectively acylated with the amino acid cognate to the original scaffold (qtRNA$^{Arg}$, qtRNA$^{Gln}$, qtRNA$^{Glu}$, qtRNA$^{Gly}$, qtRNA$^{Phe}$, qtRNA$^{Pro}$, qtRNA$^{Ser}$, qtRNA$^{Tyr}$). Four qtRNAs selectively incorporate Arg, rather than the amino acid cognate to the scaffold (qtRNA$^{Asp}$, qtRNA$^{Ile}$, qtRNA$^{Trp}$, qtRNA$^{Val}$). Two qtRNAs that are charged with the cognate amino acid as well as promiscuously charged with Arg (qtRNA$^{Ala}$ and qtRNA$^{Met}$). All 14 of these qtRNAs might be selected by a natural system to rescue a frameshift mutation. The 12 qtRNAs

that are selectively charged might be further selected as components of an all-quadruplet genetic code.

These data highlight the plasticity of AARS recognition for altered codons, both in nucleobase composition and size. The presence of a quadruplet anticodon presumably distorts the structure of the anticodon binding domain, which is a major identity element for many AARSs (*Giegé et al., 1998*). How much distortion will the enzyme functionally accept, and/or will recognition of the additional base lead to mischarging with a different amino acid? Of the eight qtRNAs that retained the cognate specificity, tRNA$^{Gly}$ and tRNA$^{Pro}$ are known as naturally occurring functional frameshift suppressors (*Riddle and Carbon, 1973*; *Sroga et al., 1992*); we confirm that they are selectively charged by the cognate AARS. Together with tRNA$^{Gln}$, our results show that *E. coli* GlnRS, GlyRS, and ProRS recognize the first three bases of the quadruplet anticodon, and are capable of charging qtRNAs despite the increased anticodon size. Correct charging of qtRNAs derived from qtRNA$^{Ser}$ is expected, as *E. coli* SerRS does not interact with the anticodon (*Biou et al., 1994*). A striking result is the correct charging of qtRNA$^{Glu}_{CGGT}$ and qtRNA$^{Phe}_{CGGC}$ by their respective *E. coli* tRNA synthetases, as their triplet anticodon sequence is unlike that of the quadruplet anticodon. However, in both cases all other known critical identity elements (*Giegé et al., 1998*) are present in the tRNA scaffold. This suggests that the major recognition feature, the anticodon, can be outweighed by the sum of the other identity elements, creating in some cases an avenue for anticodon evolution.

Finally, why did we observe acylation of multiple qtRNAs with arginine? ArgRS is responsible for synthesis of a family of Arg-tRNAs needed for the recognition of six codons, including tRNAs that differ at both the first and last position of the anticodon. As a consequence, the identity of just one anticodon position is invariant (C35). Examination of the qtRNAs charged with Arg (*Figure 5B*) shows that all satisfy this anticodon identity element, causing widespread promiscuous charging with Arg, even for qtRNAs that lack other known argRS identity elements such as A20, which is absent in qtRNA$^{Asp}_{CGGC}$ and qtRNA$^{Val}_{CGGC}$. For this reason, the presence of promiscuous ArgRS substantially increases the probability that tRNA point insertions in diverse scaffolds will result in an aminoacylated qtRNA.

Taken together, we found that in the majority of cases, qtRNAs exhibit properties that would render them evolutionarily favored building blocks: they are selectively charged by a single amino acid and incorporate that amino acid in response to their quadruplet codon.

## Compiled trends in nascent qtRNA evolution

In total, we characterized 116 different qtRNAs based on 20 tRNA scaffolds that decode 20 unique quadruplet codons (*Figure 6A*). This greatly expands the total number of known qtRNAs, the diversity of triplet tRNA scaffolds they are based upon, and the diversity of quadruplet codons they recognize beyond what has previously been reported. We found that 60 out of 109 are functional, that is, they generate increased luminescence upon induction of qtRNA expression. In total, every tRNA scaffold we tested is capable of supporting quadruplet codon translation given an appropriate four-base codon choice (i.e., in *Figure 6A* each column has at least one filled square).

These compiled trends can be interpreted in the context of known AARS identity elements (*Giegé et al., 1998*). Most prominently, we tested several qtRNAs that compete with the arginine codons AGG and found 18/20 to be functional. Further, we characterized the amino acid incorporated during translation and found that in 4/4 cases arginine is the dominant amino acid inserted. Together, these data indicate that the anticodon is a strong identity element for ArgRS, which charges tRNAs with arginine-like anticodons regardless of matching of other putative identity elements like A20 or A/G73. We also found that several isoacceptor classes were especially resistant to accepting a quadruplet anticodon. For example, 9/10 qtRNA$^{Asn}$ species tested were nonfunctional. Only qtRNA$^{Asn}_{AGGA}$ exhibits weak activity and is likely charged with arginine. This suggests that AsnRS does not tolerate expanded anticodons. Similarly, qtRNA$^{Ile}$ and qtRNA$^{Met}$ only tolerate arginine-like anticodons. Interestingly, we also found that certain isoacceptor classes do not tolerate new anticodons even when this would not be predicted from known identity elements. For example, qtRNA$^{Ser}_{TCGG}$ and qtRNA$^{Ser}_{GCAT}$ are both nonfunctional, even though SerRS does not interact with the anticodon loop of its cognate tRNA. The function of these qtRNAs may instead be governed by interaction with other enzymes that add chemical modifications.

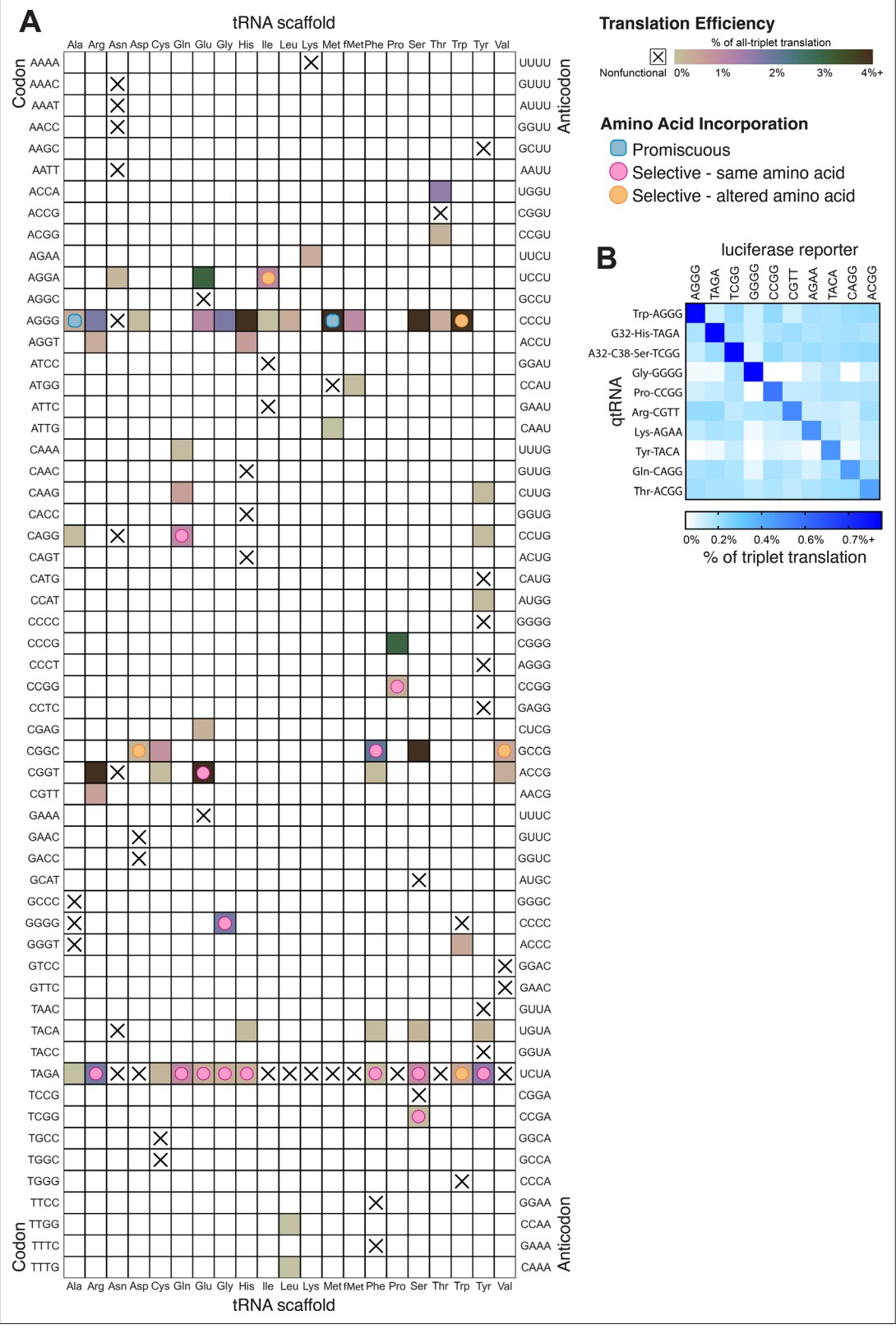

**Figure 6.** Compiled qtRNA measurements. (**A**) Compiled results of quantifying quadruplet tRNAs (qtRNA) translation efficiency (Materials and methods, 'Luciferase readthrough assay') and charging (Materials and methods, 'Quantification of qtRNA charging using mass spectrometry'). The qtRNA scaffold (columns) and codon (rows) are indicated. The translation efficiency is indicated on each qtRNA that could be measured, or marked as 'nonfunctional', indicating that the qtRNA exhibits too strong a growth defect to measure, or does not produce a measurable increase luminescence upon induction

*Figure 6 continued on next page*

*Figure 6 continued*

of qtRNA expression. Translation efficiency is measured as a percent of all-triplet translation (Materials and methods, 'Luciferase readthrough assay'). fMet qtRNAs are measured with a luciferase reporter bearing a quadruplet codon at residue 1; all others are measured with a quadruplet codon at residue 357 of luxAB. Charging results are indicated on the table for qtRNAs derived from each scaffold-codon pair; the measured qtRNAs may contain additional mutations; see *Supplementary files 2 and 3*. Data represent the mean of three to eight technical replicates in each of between one and eight biological replicates. For complete compiled raw data, see *Figure 6—source data 1*. (**B**) A miniature all-quadruplet genetic code. For each of the 10 qtRNAs (rows), we measured readthrough of a luciferase transcript containing the indicated quadruplet codon at residue 357 (columns). For raw data, see *Figure 6—source data 2*.

The online version of this article includes the following source data for figure 6:

**Source data 1.** Luminescence measurements of qtRNAs.

**Source data 2.** Luminescence measurements of qtRNA crosstalk.

To be used as part of a genetic code, qtRNAs must be capable of operating together with minimal crosstalk. We selected 10 qtRNAs and measured crosstalk between members of this set, revealing a high degree of orthogonality: each qtRNA is specific for its own four-base codon (*Figure 6B*).

## Discussion

We present a comprehensive study of whether tRNAs can be altered to functionally decode quadruplet codons. Our results show that quadruplet decoding is a remarkably universal phenomenon: every tRNA we tested is capable of exhibiting weak quadruplet decoding activity with four or fewer mutations. Additionally, weak quadruplet decoding requires very few mutations in many cases. For example, a single point insertion in the anticodon is sufficient to generate functional qtRNAs for 7/20 isoacceptor classes.

The tRNA anticodon is a major recognition element for many tRNA synthetases (*Giegé et al., 1998*). Thus we were surprised to find nine different AARSs (ArgRS, GluRS, GlnRS, GlyRS, HisRS, PheRS, ProRS, SerRS, TyrRS) that selectively charged the cognate qtRNA, creating the opportunity for amino acid and codon-diverse all-quadruplet peptide translation. The diversity of codons that can be recognized by qtRNAs can easily enable codon-selective translation, as we demonstrate with a miniature all-quadruplet genetic code composed of 10 orthogonal qtRNAs based on unique isoacceptor classes. Together, these results demonstrate that qtRNAs capable of amino acid-selective efficient translation of quadruplet codons can arise during evolution.

However, our study identifies several factors that may have prevented universal adoption of an all-quadruplet code. First, we find that some qtRNAs exhibit strong fitness defects which prevent them from being maintained and accumulated in genomes. Second, additional mutations outside the anticodon are often required for efficient translation, reducing the probability that efficient qtRNAs would arise by chance. Such mutations may affect identity elements of the tRNA modifying enzymes that chemically alter the bases surrounding the anticodon (*Hori, 2014*); similar patterns have been observed for canonical triplet tRNAs (*Yarus, 1982*). Finally, only some AARSs are amenable to altered codon sizes for some amino acids: 12/20 modern *E. coli* AARSs are intolerant, inefficient, or promiscuous with qtRNAs. In particular, AARSs that arose later during evolution may be more finely tuned to precise anticodon base recognition and thus less tolerant of anticodon mutations or expansion. A nascent all-quadruplet code would initially have access to only the limited chemical vocabulary of the nine AARSs that are most amenable to quadruplet codon translation, decreasing the likelihood that valuable quadruplet-ORFs might arise naturally and be selected. Similarly, as evolution progressed, the translation apparatus would have been under increasingly strong selection favoring increased fidelity of translation to support a genetic code with an increased number of codons, making it increasingly difficult for an all-quadruplet codon code to arise. Finally, studies have shown that ORFs can also arise de novo from non-coding regions (*Carvunis et al., 2012*) and that random ORFs can rapidly evolve into functional proteins (*Keefe and Szostak, 2001*), offering an avenue for gene birth in a different codon size should the appropriate qtRNAs be available. However, given that the canonical genetic code only includes 20 or 21 amino acids, even though there are 64 codons, suggests that there is little natural selective pressure to expand the genetic code to a much larger number than the currently known canonical amino acids. Together, these factors contribute to the absence of naturally occurring quadruplet codon codes, despite the frequency with which individual quadruplet translation

components can arise through evolution. We expect that naturally occurring instances of qtRNAs or quadruplet codon codes would arise even less frequently than other non-canonical genetic codes in bacteria (*Shulgina and Eddy, 2021*).

Although these factors explain the absence of naturally occurring all-quadruplet genetic codes, our results offer a blueprint for deliberately engineering an expanded all-quadruplet genetic code. While there is much current interest in the detailed aspects of quadruplet decoding on the ribosome (*Gamper et al., 2021*; *Choi et al., 2020*), more work is needed to understand the hurdles a mRNA quadruplet codon encounters on its way through the ribosome. In addition, tRNA modification of the anticodon loop may be crucial to endow the anticodon with the flexibility needed to efficiently pair with the quadruplet codon in the ribosome environment (*Grosjean and Westhof, 2016*). Our work was not designed to contribute to mechanistic understanding. Instead, we aim to survey the empirical properties of qtRNAs. Our data have revealed exactly which triplet codons can be reassigned to quadruplet codons without compromising amino acid selectivity: an initial all-quadruplet code will be composed of nine selectively charged qtRNAs from this study (qtRNA$^{Gln}_{CAGG}$, qtRNA$^{Pro}_{CCGG}$, qtRNA$^{Phe}_{CGGC}$, qtRNA$^{Glu}_{CGGT}$, qtRNA$^{Gly}_{GGGG}$, qtRNA$^{Ser}_{TCGG}$, qtRNA$^{Arg}_{AGGG}$, and either qtRNA$^{Tyr}_{TAGA}$ or qtRNA$^{His}_{TAGA}$).

Further, the flexibility of AARSs suggests that anticodon re-engineering may be a viable strategy for reassignment of the additional 11 canonical amino acids to quadruplet codons. We and others (*Krishnakumar et al., 2013*) have observed that ArgRS is a major source of promiscuous charging; refactoring or replacing this ArgRS may be broadly valuable for enabling reassignment of arginine-like codons; that is, enabling selective charging of qtRNA$^{Ala}_{AGGG}$. A primary limitation of quadruplet codon translation is competition with the canonical triplet tRNAs, resulting in toxicity and low translation efficiency. To overcome this major limitation, strategies such as CCA tail engineering (*Samaha et al., 1995*; *Terasaka et al., 2014*), or ribosome engineering (*Neumann et al., 2010*; *Schmied et al., 2018*) to reject triplet codons, will be essential for creating an all-quadruplet genetic code. Additionally, current qtRNAs are known to crosstalk with four-base mismatched codons (*DeBenedictis et al., 2021*), and engineering to eliminate this behavior will be required to fully utilize the expanded codon set. Re-engineering the majority of endogenous tRNA synthetases is a considerable engineering investment, but advances in protein engineering likely make this task not more onerous than current genetic code expansion techniques, which require years of iterative whole genome synthesis and assembly to free up a large number of codons. Together, our deliberate exploration of the evolution of functional quadruplet translation will launch synthetic efforts to assemble a 256 codon genetic code.

# Materials and methods

**Key resources table**

| Reagent type (species) or resource | Designation | Source or reference | Identifiers | Additional information |
|---|---|---|---|---|
| Strain, strain background (*Escherichia coli*) | S2060 | Addgene | #105064 | K12 derivative optimized for directed evolution |
| Recombinant DNA reagent | pED1a6 (plasmid) | Addgene | #134787 | Luciferase reporter for quadruplet codon translation |
| Recombinant DNA reagent | pED17 × 11 (plasmid) | Addgene | #134814 | sfGFP reporter for quadruplet codon translation |
| Recombinant DNA reagent | pED7 × 10 (plasmid) | Addgene | #134812 | pIII reporter for quadruplet codon translation |
| Recombinant DNA reagent | pED14xS-1(plasmid) | Addgene | #134800 | IPTG inducible expression of qtRNA |

## General methods
### Antibiotics
Antibiotics (Gold Biotechnology) were used at the following working concentrations: carbenicillin, 50 µg/mL; spectinomycin, 100 µg/mL; chloramphenicol, 40 µg/mL; kanamycin, 30 µg/mL; tetracycline, 10 µg/mL; streptomycin, 50 µg/mL.

## Media
David Rich Medium (DRM) (*Bryson et al., 2017*) (US Biological, #CS050H-001 and #CS050H-003) is used for luminescence assays due to its low fluorescence and luminescence background. 2XYT media

(US Biological, T9200), a media optimized for phage growth, is used for all other purposes, including phage-based selection assays and general cloning. Agar (US Biological, A0930) is used for cloning and plaque assays.

## Plasmid cloning

Plasmids were cloned using either Mach1, Turbo, DH5a, or 10beta cells. Unless otherwise noted, plasmids and phage were cloned by USER assembly (*Geu-Flores et al., 2007*) using the Phusion U Hot Start DNA polymerase (Thermofisher, F556L) and USER enzyme (New England Biolabs, M5505L).

## Preparation of chemically competent cells

Strain S2060 (Addgene #105064), a K12 derivative optimized for directed evolution (*Hubbard et al., 2015*) was used in all luciferase, phage propagation, and plaque assays. To prepare competent cells, an overnight culture was diluted 1000-fold into 50 mL of 2XYT media supplemented with maintenance antibiotics and grown at 37°C with shaking at 230 rpm to $OD_{600}$ ~0.4–0.6. Cells were pelleted by centrifugation at 6000× $g$ for 10 min at 4°C. The cell pellet was then resuspended by gentle stirring in 5 mL of TSS (LB media supplemented with 5% v/v DMSO, 10% w/v PEG 3350, and 20 mM $MgCl_2$). The cell suspension was stirred to mix completely, aliquoted and flash-frozen in liquid nitrogen, and stored at −80°C until use.

## Transformation of chemically competent cells

To transform cells, 100 µL of competent cells were thawed on ice. To this, plasmid (2 µL each of miniprep-quality plasmid; up to two plasmids per transformation) and 100 µL KCM solution (100 mM KCl, 30 mM $CaCl_2$, and 50 mM $MgCl_2$ in $H_2O$) were added and stirred gently with a pipette tip. The mixture was incubated on ice for 10 min and heat shocked at 42°C for 45 s. The mixture was chilled on ice for 4 min, then 850 µL of 2XYT media was added. Cells were allowed to recover at 37°C with shaking at 230 rpm for 0.75 hr, streaked on 2XYT media + 1.5% agar plates containing the appropriate antibiotics, and incubated at 37°C for 16–18 hr.

## Standard phage cloning

Competent *E. coli* S2060 cells were prepared containing pJC175e (Addgene #79219), a plasmid expressing pIII under control of the phage shock promoter (*Badran et al., 2016*), which enables propagation of ΔpIII M13 bacteriophage through complementation. To clone ΔpIII M13 bacteriophage, PCR fragments were assembled using USER. The annealed fragments were transformed into competent S2060-pJC175e competent cells. Transformants were recovered in 2XYT media overnight, shaking at 230 rpm at 37°C. The phage supernatant from the resulting culture was filtered through a 0.22 µm membrane (Thomas Scientific, 1166 U41), and plaqued to isolate clonal phage (see below). Clonal plaques were picked into 2 mL of 2XYT media and expanded overnight, shaking at 230 rpm at 37°C, filtered, and Sanger sequenced.

### tRNA diagrams

R2R was used to generate tRNA diagrams (*Weinberg and Breaker, 2011*). R2R is free software available from http://www.bioinf.uni-leipzig.de/~zasha/R2R/.

## tRNA scaffold selection

We used the first scaffold listed in each isoacceptor class, based on ecogene.org listing for *E. coli* K12 circa January 2019. tRNA genes were amplified directly from *E. coli* genomic DNA; see *Supplementary file 1* for tRNA sequences and links to Benchling plasmid maps.

## Phage plaque assays

### Manual protocol

S2060 cells were transformed with the Accessory Plasmid of interest. Overnight cultures of single colonies grown in 2XYT media supplemented with maintenance antibiotics were diluted 1000-fold into fresh 2XYT media with maintenance antibiotics and grown at 37°C with shaking at 230 rpm to $OD_{600}$ ~0.6–0.8 before use. Bacteriophage were serially diluted 100-fold (four dilutions total) in $H_2O$;

100 µL of cells were added to 100 µL of each phage dilution, and to this 0.85 mL of liquid (70°C) top agar (2XYT media + 0.6% agar) supplemented with 2% Bluo-Gal (GoldBio, CAS #97753-82-7) was added and mixed by pipetting up and down once. This mixture was then immediately pipetted onto one quadrant of a quartered petri dish (VWR, 25384308) already containing 2 mL of solidified bottom agar (2XYT media + 1.5% agar, no antibiotics). After solidification of the top agar, plates were incubated at 37°C for 16–18 hr.

## Robotics-accelerated protocol

Plaque assays were automated as previously described (*Chory et al., 2021*). Briefly, the same procedure was followed as above, except that plating of the plaque assays was done by a liquid handling robot (Hamilton Robotics) by plating 20 µL of bacterial culture and 100 µL of phage dilution with 200 µL of soft agar onto a well of a 24-well plate already containing 235 µL of hard agar per well. To prevent premature cooling of soft agar, the soft agar was placed on the deck in a 70°C heat block.

## Luciferase readthrough assay

Luciferase readthrough assays were performed as previously described (*DeBenedictis et al., 2021*). Briefly, S2060 bacteria were transformed with a luciferase reporter containing a quadruplet codon at permissive serine residue 357 (*DeBenedictis et al., 2021*) (e.g., https://benchling.com/s/seq-7CsWcP8Ez4JNjNM9W23N, Addgene #134787) and an inducible qtRNA expression plasmid (e.g., https://benchling.com/s/seq-F5NZDNmxOhUoWp5DX41h, Addgene #134800). Bacteria were grown overnight, then diluted 500-fold the next day into DRM with qtRNA expression induced (1 mM IPTG, GoldBio, I2481C5), or suppressed (0 mM IPTG). Absorbance and luminescence measurements are taken kinetically in a ClarioSTAR plate reader (BMG Labtech) over the course of 8 hr. Each biological replicate is taken with three to eight technical replicates, each of which corresponds to a unique colony that is picked. To account for differential growth rate, all luminescence values are considered at $OD_{600} = 0.5$. The % of triplet translation efficiency is calculated using the formula (QuadLux $_{qtRNA\ induced}$ – QuadLux $_{qtRNA\ uninduced}$)/(TriLux – QuadLux $_{qtRNA\ uninduced}$), where:

> TriLux is the luminescence of the positive control, a luciferase encoded entirely with triplet codons.
> QuadLux qtRNA induced is the luminescence produced by the quadruplet codon-bearing reporter upon qtRNA expression (1 mM IPTG).
> QuadLux qtRNA uninduced is the luminescence produced by the quadruplet codon-bearing reporter in the absence of qtRNA expression (0 mM IPTG).

Note that though kinetic measurement assists in accounting for differential growth rate, it remains the case that bacteria may exhibit growth differences that cannot be fully compensated for by kinetic measurement. Negative % translation efficiency values may be attributed to the effects of toxicity.

## Growth defect

Our measurement of growth defect involves analysis of the absorbance measured during the 8 hr growth curves taken in the luciferase readthrough assay. We identify the time, $t_{measure}$ at which QuadLux $_{qtRNA\ induced}$ reaches $OD_{600} = 0.5$. Growth defect = (OD of QuadLux $_{qtRNA\ induced}$ at $t_{measure}$)/(OD of QuadLux $_{qtRNA\ suppressed}$ at $t_{measure}$).

## Phage library primer design

We do not recommend USER cloning for library creation inside of high-secondary structure tRNAs; instead, we used degenerate primers and blunt end ligation. Primers were designed for use with around-the-world PCR, creating a one-piece blunt end ligation. In order to reduce nucleotide bias during blunt end ligation assembly, the last degenerate base was designed to be at least one base away from the end of the primer. In all cases, phage bearing the wildtype triplet tRNA scaffold are used as a template for the PCR in order to eliminate background bias in the libraries.

## Degenerate NNNN anticodon libraries

Degenerate nucleotides are included at bases 34, 35, 35.5, and 36 of the qtRNA. See an example of this primer design: https://benchling.com/s/seq-5k2FZdPbWpiB8tOh8WCK.

## Primer design library diversified at positions 32, 37, and 38 of qtRNA

Degenerate nucleotides are included at bases 32, 37, and 38 of the qtRNA. See an example of this primer design: https://benchling.com/s/seq-oMMGFRl2vITYpNX2HWkz.

## Phage library cloning

Primers for around-the-world PCR and blunt end ligation were designed as described above. For each library, 200 µL of PCR product was produced using Phusion Hotstart Flex polymerase (New England Biolabs, M0535S). The entirety of this PCR product was run on a gel, extracted, and purified using spin column purification (Qiagen, 28106). Background plasmid was digested using Dpn1 (New England Biolabs, R0176L), and the remaining PCR product was purified again using spin columns, and ligated. The ligation product was transformed into competent *E. coli* S2060 cells containing pJC175e (as described). Transformants were recovered in 2XYT media overnight, shaking at 230 rpm at 37°C. The phage supernatant from the resulting culture was filtered through a 0.22 µm membrane (Thomas Scientific, 1166 U41), and plaqued (as described).

## pIII-based selection of NNNN anticodon libraries

We selected eight quadruplet codons of interest, focusing on codons for which at least one functional qtRNA was already known that could act as positive control. In total we selected six XYZZ codons (CGTT, GGGG, CCGG, CAGG, TCGG, ACGG) and two arginine-like codons (AGGG, CGGT). To perform selection, competent S2060 cells were transformed with a plasmid expressing the appropriate quadruplet codon replacing permissive proline residue pIII-29 (*Bryson et al., 2017*). A picked colony was grown overnight in 2XYT media with maintenance antibiotics. Following overnight growth, cultures were diluted 1:1000 in 2XYT media without antibiotics and 500 µL of diluted culture was aliquoted into a 2 mL deep 96-well plate. Wells were inoculated with phage encoding the desired qtRNA library to a final concentration of $10^5$ pfu/mL. The plate was grown overnight, shaking at 230 rpm at 37°C, and then the phage supernatant was filtered and plaqued in activity-independent host S2060 cells bearing pJC175e. Selections were ranked from lowest final phage titer to highest final phage titer. Two plaques per selection were picked for Sanger sequencing in all selections that enrich over 10-fold. For isoacceptor classes for which nothing enriched over 10-fold, additional plaques were picked, beginning with the most-enriched.

## pIII-based selection and NGS of libraries diversified at positions 32, 37, and 38

We picked 41 functional qtRNAs, including at least one qtRNA for every unique scaffold, and picking qtRNAs with the highest-function first. Competent S2060 cells were transformed with a plasmid expressing the appropriate quadruplet codon replacing permissive residue pIII-29 (*Bryson et al., 2017*). A picked colony was grown overnight in 2XYT media with maintenance antibiotics. Following overnight growth, cultures were diluted 1:1000 in 2XYT media without antibiotics and 500 µL of diluted culture was aliquoted into a 2 mL deep 96-well plate. Wells were inoculated with phage encoding the desired qtRNA library to a final concentration of $10^5$ pfu/mL. The plate was grown overnight, shaking at 230 rpm at 37°C, and then the phage supernatant was filtered and plaqued in activity-independent host S2060 cells bearing pJC175e. Libraries were amplified using PCR and sequenced on a MiSeq to >10× library size.

## Quantification of qtRNA charging using mass spectrometry

Each qtRNA (e.g., https://benchling.com/s/seq-BpqbupgoHvZu0gFmNUBm) was co-expressed with C-terminal 6xHis-tagged *sfGFP* (e.g., https://benchling.com/s/seq-bI1bixktGKegGwboMYIP) with the appropriate quadruplet codon replacing permissive tyrosine residue sfGFP-151 *Young et al., 2010* in S2060 cells. GFP was purified from these cultures at either 4 mL or 1 L scale (*Supplementary file 2*).

## Purification at 4 mL scale

Co-transform bacteria with sfGFP-151-quad and qtRNA expression plasmids. Inoculate a single colony of the expression strain in DRM (with 100 µg/mL spectinomycin, 30 µg/mL kanamycin, and 25 mM glucose) as the seed culture; grow overnight at 37°C, 200 rpm. Dilute 1:1000 and into DRM

that induces qtRNA expression (with 100 µg/mL spectinomycin, 30 µg/mL kanamycin, and 1 mM IPTG) and grow for 29 hr, 200 rpm, at 37°C. Cultures were spun down at 5000 $g$ for 10 min and the pellet frozen at –80°C. Thawed pellets were lysed using B-PER Complete Reagent (Thermofisher 89821) and His-tagged protein was purified from cell lysate using a Ni-NTA spin column (Qiagen, 31014). The resulting product was run on a 12% Bis-Tris PAGE gel (Thermofisher, NP0342PK2), and the appropriate band was extracted for mass spectrometry analysis.

### Purification at 1 L scale

Proteins were purified by BiologicsCorp at 1 L scale. Inoculate a single colony of the expression strain in LB media (with 100 µg/mL spectinomycin, 30 µg/mL kanamycin, and 25 mM glucose) as the seed culture; at 37°C, 200 rpm, overnight. Perform 1% inoculation into 500 mL TB media (with 100 µg/mL spectinomycin, 30 µg/mL kanamycin, and 25 mM glucose); at 37°C, 200 rpm, till $OD_{600}$ = 0.6–0.8. Add IPTG to a final concentration of 1 mM. Continue culturing for 4 hr. Collect cells by centrifugation (4°C, 8,000 rpm, 15 min). Wash cells once by resuspension with PBS buffer and again collect cells by centrifugation. Resuspend per gram of wet cell pellet with 20 mL Lysis Buffer T300-20L (50 mM Tris-HCl pH 8.0, 300 mM NaCl, and 20 mM imidazol, plus 1 mM DTT, 1% Triton X-114, 1 µg/mL Pepstatin A, and 1 µg/mL Leupeptin). Sonicate the cell suspension on ice for 15–25 min (500 W; 3 s burst and 6 s pause). Centrifuge the cell lysate; at 4°C, 12,500 rpm, for 15 min. Filter the supernatant through a 0.45 µm pore membrane. Purify His-tagged protein using a Ni-IDA column. Analyze fractions by SDS-PAGE. For samples with satisfactory purity, pool fractions and dialyze against PBS buffer (pH 7.4); at 4°C; 2 hr per cycle, 3–4 cycles. Filter the protein solution through a 0.22 µm pore membrane. Concentrate if necessary.

### Mass spectrometry analysis

Samples were trypsin digested and measured using HPLC-MS/MS. To analyze the results, the resulting fragmentation spectra were correlated against a custom database containing the predicted fragmentation pattern for the fragment straddling residues 151 if it contained each of the 20 canonical amino acids. Abundance of each species was quantified by calculating the area under the curve of the ion chromatogram for each peptide precursor (*Supplementary file 3*). The limit of detection was $10^4$ [AU], the lower limit for area under the curve for a peptide on this instrument. The mass spectrometry proteomics data have been deposited to the ProteomeXchange Consortium via the PRIDE (*Perez-Riverol et al., 2022*) partner repository with the dataset identifier PXD031925 and 10.6019/PXD031925.

## Data and materials availability

Key plasmids from this study have been deposited on Addgene. Raw spectra have been deposited in the PRIDE database (*Perez-Riverol et al., 2022*), dataset identifier PXD031925 and 10.6019/PXD031925.

## Acknowledgements

We thank Eric Spooner and the Whitehead Proteomics Core Facility. We thank the laboratory of Kristala Prather for equipment use and assistance. We thank Nili Ostrov, Ben Thuroni, Stephen Von Stetina, and Sergey Melnikov for their helpful comments on the manuscript. We thank BiologicsCorp for protein purification. Funding: This work was supported by the MIT Media Lab, an Alfred P Sloan Research Fellowship (to KME), gifts from the Open Philanthropy Project and the Reid Hoffman Foundation (to KME), and the National Institute of Digestive and Kidney Diseases (R00 DK102669-01 to KME), and by the National Institute of General Medical Sciences (R35GM122560 and 3R35GM122560-05W1 to DS). EAD was supported by the National Institute for Allergy and Infectious Diseases (F31 AI145181-01).

## Additional information

### Competing interests

Erika Alden DeBenedictis, Kevin M Esvelt: filed US Patent 16405380 on tRNA sequences engineered in this work. The other author declares that no competing interests exist.

## Funding

| Funder | Grant reference number | Author |
| --- | --- | --- |
| National Institute of General Medical Sciences | R35GM122560 | Dieter Söll |
| National Institute of General Medical Sciences | 3R35GM122560-05W1 | Dieter Söll |
| National Institute of Allergy and Infectious Diseases | F31 AI145181-01 | Erika Alden DeBenedictis |
| National Institute of Diabetes and Digestive and Kidney Diseases | R00 DK102669-01 | Kevin M Esvelt |
| MIT Media Lab | | Kevin M Esvelt |
| Alfred P Sloan Research Fellowship | | Kevin M Esvelt |
| Open Philanthropy Project | | Kevin M Esvelt |
| Reid Hoffman Foundation | | Kevin M Esvelt |

The funders had no role in study design, data collection and interpretation, or the decision to submit the work for publication.

## Author contributions

Erika Alden DeBenedictis, Conceptualization, Data curation, Formal analysis, Investigation, Methodology, Writing – original draft, Writing – review and editing; Dieter Söll, D.S. Interpreted some of the experiments, Supervision, Writing – review and editing; Kevin M Esvelt, Funding acquisition, Supervision, Writing – review and editing

## Author ORCIDs

Erika Alden DeBenedictis  http://orcid.org/0000-0002-7933-2651
Dieter Söll  http://orcid.org/0000-0002-3077-8986
Kevin M Esvelt  http://orcid.org/0000-0001-8797-3945

## Decision letter and Author response

Decision letter https://doi.org/10.7554/eLife.76941.sa1
Author response https://doi.org/10.7554/eLife.76941.sa2

# Additional files

## Supplementary files

• Supplementary file 1. Table of tRNA scaffolds. Canonical triplet tRNAs were cloned from *Escherichia coli* K12 genomic DNA. Body text *Figures 2 and 3* concern mutations to the anticodon (bolded caps); body text *Figure 4* concerns mutation to positions 32, 37, and 38 (bolded lower case). We use qtRNA$^\text{three letter scaffold}_\text{four letter DNA codon}$ nomenclature to refer to qtRNAs; for example, to create qtRNA$^\text{Ser}_\text{TAGA}$, the CGA anticodon in tRNA$^\text{Ser}$ scaffold would be replaced by TCTA.

• Supplementary file 2. Purification scale of qtRNA-translated proteins. For sfGFP purification, qtRNA expression plasmids (as listed) were co-expressed with C-terminal 6xHis-tagged *sfGFP* with the appropriate quadruplet codon replacing permissive residue 151. For example, sfGFP-151-GGGG, https://benchling.com/s/seq-bl1bixktGKegGwboMYIP. Peptides detected are listed in *Supplementary file 3*. Raw spectra have been deposited in the PRIDE database (*Perez-Riverol et al., 2022*), dataset identifier PXD031925 and 10.6019/PXD031925.

• Supplementary file 3. Mass spectrometry of peptides flanking residue 151. List of peptides flanking residue 151 that were detected during mass spectrometry. Raw spectra have been deposited in the PRIDE database (*Perez-Riverol et al., 2022*), dataset identifier PXD031925 and 10.6019/PXD031925.

• Transparent reporting form

## Data availability

All luminescence raw data are compiled in Figure 6A and provided as Source Data 1. Raw mass spectometry spectra have been deposited in the PRIDE database, dataset identifier PXD031925 and https://doi.org/10.6019/PXD031925.

The following dataset was generated:

| Author(s) | Year | Dataset title | Dataset URL | Database and Identifier |
|---|---|---|---|---|
| DeBenedictis EA, Esvelt K | 2022 | Measurement of the amino acid incorporation of tRNAs engineered to decode four-base codons | https://www.ebi.ac.uk/pride/archive/projects/PXD031925 | PRIDE, PXD031925 |

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
