## [Editor Report]

Using a phage-based library generation and selection, the authors generated a suite of 4-base decoding tRNAs with improved efficiency in quadruplet decoding. The data represent an important step toward enhancing protein synthesis with 4-base codons. Overall, the approach to generate many tRNA variants with quadruplet anticodons is intriguing and provides a wealth of valuable information to the field. The results should become foundational for the field of synthetic biology.

---

## [Decision Letter]

**Decision letter after peer review:**

Thank you for submitting your article "Measuring the tolerance of the genetic code to altered codon size" for consideration by *eLife*. Your article has been reviewed by 3 peer reviewers, and the evaluation has been overseen by Timothy Nilsen as the Reviewing Editor and James Manley as the Senior Editor. The following individuals involved in review of your submission have agreed to reveal their identity: Gerald F Joyce (Reviewer #3).

All of the reviewers were quite positive about the work, but each has made several suggestions for improvements of the manuscript (see reviews below). It is particularly important that you provide a clearer distinction between what you hope to achieve in the future and what you have actually achieved in this study. It was agreed that multiplexing with several 4-based codons will be challenging especially when competing with 3-base decoding. The efficiency of 4-base decoding is very low even with some improvements through directed evolution.

We believe that the main message is that the genetic code is surprisingly tolerant to quadruplet codons across the board. We do want to see the primary mass spec data rather than a bar graph summary. This data could be deposited in PRIDE or similar databases. It is a requirement for publication in *eLife* that these be provided as extended data. We also suggest that a discussion of what needs to be improved, specific aminoacylation, better handling of decoding, faster translocation, etc., should be included in the paper to achieve the dream of 256 codons. Please address these points as thoroughly as possible and other points that are raised in the reviews that you feel that you can address.

*Reviewer #1 (Recommendations for the authors):*

1. Abstract: ' These constitute a functional and mutually orthogonal set' unclear how the quadruplet de-coding tRNAs will be orthogonal, since they all compete with 3-base decoding.

2. Intro: ' including hundreds of free codons that could be assigned to noncanonical amino acids' this is not correct, none of them are open as all of these codons would compete with triplet de-coding of normal reading frames.

3. 'faithfully and efficiently charge a cognate qtRNA' not support by the data.

4. 'In total, every tRNA scaffold we tested is capable of supporting quadruplet codon translation given appropriate four-base codon choice' Not support by the data. I believe some showed luminescence indistinguishable from background.

5. ' other point insertions create qtRNAs that do not functionally decode quadruplet codons' would a different codon context show some of these tRNAs to be functional?

6. 'functional quadruplet translation will launch synthetic efforts to assemble a 256-amino acid genetic code.' 256 codons does not provide 256 AAs, what will be the redundancy of the quadruplet code? An expanded discussion on this point would improve the manuscript.

*Reviewer #3 (Recommendations for the authors):*

1. Please avoid self-aggrandizing statements such as in the first sentence of the Discussion: "We present the first-ever comprehensive study…" (change to "We present a comprehensive study…"; or in the third sentence of the last paragraph of the Discussion: "Our data has (have) revealed, for the first time… (delete "for the first time").

2. Source data for the mass spec analyses need to be provided.

---

## [Author Response]

All of the reviewers were quite positive about the work, but each has made several suggestions for improvements of the manuscript (see reviews below). It is particularly important that you provide a clearer distinction between what you hope to achieve in the future and what you have actually achieved in this study. It was agreed that multiplexing with several 4-based codons will be challenging especially when competing with 3-base decoding. The efficiency of 4-base decoding is very low even with some improvements through directed evolution.We believe that the main message is that the genetic code is surprisingly tolerant to quadruplet codons across the board. We do want to see the primary mass spec data rather than a bar graph summary. This data could be deposited in PRIDE or similar databases. It is a requirement for publication in eLife that these be provided as extended data. We also suggest that a discussion of what needs to be improved, specific aminoacylation, better handling of decoding, faster translocation, etc., should be included in the paper to achieve the dream of 256 codons. Please address these points as thoroughly as possible and other points that are raised in the reviews that you feel that you can address.

We thank you for conducting this review. We have deposited the raw mass spectrometry data in the PRIDE database, and have also provided the peptide fragmentation data in Supplementary File 3. We have expanded upon the discussion to more fully explore future engineering steps that would be necessary to achieve an all-quadruplet codon genetic code.

Reviewer #1 (Recommendations for the authors):1. Abstract: ' These constitute a functional and mutually orthogonal set' unclear how the quadruplet de-coding tRNAs will be orthogonal, since they all compete with 3-base decoding2. Intro: ' including hundreds of free codons that could be assigned to noncanonical amino acids' this is not correct, none of them are open as all of these codons would compete with triplet de-coding of normal reading frames.

In the abstract and introduction we are referencing a concept that has been popularized in the literature (i.e. Reprogramming the genetic code, Chin, 2021) that an all-quadruplet codon code could be created that is selective for quadruplet codons. We agree that the literature has been light on a clear discussion of exactly what would be necessary to achieve an all-quadruplet codon code. In this manuscript, we tackle the question, how do you incorporate all 20 canonical amino acids?

Competition with triplet codons is certainly a second big hurdle. Mechanistically, this might be achieved in several ways, including evolving the ribosome to reject triplet codons, or creating a separate pool of ribosomes and tRNAs using CCA tail engineering. We have added this to the discussion:

“A primary limitation of quadruplet codon translation is competition with the canonical triplet tRNAs, resulting in toxicity and low translation efficiency. To overcome this major limitation, strategies such as CCA tail engineering, or ribosome engineering to reject triplet codons, will be essential for creating an all-quadruplet genetic code.”

3. 'faithfully and efficiently charge a cognate qtRNA' not support by the data.

Thank you for the comment, we have removed this.

4. 'In total, every tRNA scaffold we tested is capable of supporting quadruplet codon translation given appropriate four-base codon choice' Not support by the data. I believe some showed luminescence indistinguishable from background.

As you point out, we tested many qtRNAs that do not exhibit luminescence that is distinguishable from background. Those “non-functional” qtRNAs are marked with an ‘X’ in Figure 6A. In addition, we tested many “functional” qtRNAs that *do* exhibit luminescence above background. They are indicated with a filled square in Figure 6A. There is at least one filled square in each column because we found at least one functional qtRNA based on a tRNA scaffold from every isoacceptor class.

The full statement in the paper is correct, “In total, every tRNA scaffold we tested is capable of supporting quadruplet codon translation given appropriate four-base codon choice (that is, in Figure 6A each column has at least one filled square).”

5. ' other point insertions create qtRNAs that do not functionally decode quadruplet codons' would a different codon context show some of these tRNAs to be functional?

Throughout the paper, we experimented with several different reporter genes that place the quadruplet codon of interest in a different mRNA context, including luxAB, sfGFP, and pIII. We found that the amount of quadruplet decoding activity was correlated between these reporters, for example, compare Figure 4D (pIII context) with Figure 4E (luxAB context). We did not happen to identify any qtRNAs that were functional in one context but not another, although we did identify qtRNAs that interact with an unexpected area of the transcript as in Figure 3F.

6. 'functional quadruplet translation will launch synthetic efforts to assemble a 256-amino acid genetic code.' 256 codons does not provide 256 AAs, what will be the redundancy of the quadruplet code? An expanded discussion on this point would improve the manuscript.

We have expanded the discussion on this point, which now reads:

“A primary limitation of quadruplet codon translation is competition with the canonical triplet tRNAs, resulting in toxicity and low translation efficiency. To overcome this major limitation, strategies such as CCA tail engineering^31,32^, or ribosome engineering^33^ to reject triplet codons, will be essential for creating an all-quadruplet genetic code. Additionally, current qtRNAs are known to crosstalk with four-base mismatched codons^9^, and engineering to eliminate this behavior will be required to fully utilize the expanded codon set.”

Reviewer #3 (Recommendations for the authors):1. Please avoid self-aggrandizing statements such as in the first sentence of the Discussion: "We present the first-ever comprehensive study…" (change to "We present a comprehensive study…"; or in the third sentence of the last paragraph of the Discussion: "Our data has (have) revealed, for the first time… (delete "for the first time").

We have removed these statements.

2. Source data for the mass spec analyses need to be provided.

Thank you for the comment. We have added Supplementary file 3, which lists peptides detected in mass spectrometry that span residue 151. We have also deposited the raw mass spectrometry data in the PRIDE database.